# From the Argonauts Mythological Sailors to the Argonautes RNA-Silencing Navigators: Their Emerging Roles in Human-Cell Pathologies

**DOI:** 10.3390/ijms21114007

**Published:** 2020-06-03

**Authors:** Vasiliki I. Pantazopoulou, Stella Georgiou, Panos Kakoulidis, Stavroula N. Giannakopoulou, Sofia Tseleni, Dimitrios J. Stravopodis, Ema Anastasiadou

**Affiliations:** 1Center of Basic Research, Biomedical Research Foundation of the Academy of Athens (BRFAA), 11527 Athens, Greece; vaspantazo@bioacademy.gr (V.I.P.); sgeorgiou@bioacademy.gr (S.G.); pkakoulidis@di.uoa.gr (P.K.); stavroulagiannakopoulou97@gmail.com (S.N.G.); 2Section of Cell Biology and Biophysics, Department of Biology, School of Science, National and Kapodistrian University of Athens (NKUA), 15701 Athens, Greece; dstravop@biol.uoa.gr; 3Department of Pathology, Medical School, National and Kapodistrian University of Athens (NKUA), 11527 Athens, Greece; stseleni@med.uoa.gr

**Keywords:** argonaute proteins, autoimmune disease, cancer, human disease, infertility, metabolic disorders, neuronal deficiencies, post-transcriptional regulation, RNA binding proteins, viral infection

## Abstract

Regulation of gene expression has emerged as a fundamental element of transcript homeostasis. Key effectors in this process are the Argonautes (AGOs), highly specialized RNA-binding proteins (RBPs) that form complexes, such as the RNA-Induced Silencing Complex (RISC). AGOs dictate post-transcriptional gene-silencing by directly loading small RNAs and repressing their mRNA targets through small RNA-sequence complementarity. The four human highly-conserved family-members (AGO1, AGO2, AGO3, and AGO4) demonstrate multi-faceted and versatile roles in transcriptome’s stability, plasticity, and functionality. The post-translational modifications of AGOs in critical amino acid residues, the nucleotide polymorphisms and mutations, and the deregulation of expression and interactions are tightly associated with aberrant activities, which are observed in a wide spectrum of pathologies. Through constantly accumulating information, the AGOs’ fundamental engagement in multiple human diseases has recently emerged. The present review examines new insights into AGO-driven pathology and AGO-deregulation patterns in a variety of diseases such as in viral infections and propagations, autoimmune diseases, cancers, metabolic deficiencies, neuronal disorders, and human infertility. Altogether, AGO seems to be a crucial contributor to pathogenesis and its targeting may serve as a novel and powerful therapeutic tool for the successful management of diverse human diseases in the clinic.

## 1. Introduction

Argonaute (AGO) proteins were initially discovered through their implication in plant development [1,2] and stem-cell division in flies [3]. During the last two decades, extensive research has been performed to identifying the AGO-dependent processes controlling cellular physiology. Their remarkable roles as essential components of the RNA-Induced Silencing Complex (RISC) were unraveled in mammals and other organisms. AGOs dictate post-transcriptional gene-silencing mechanism(s) by repressing the mRNA targets through small RNA-sequence complementarity with or without mRNA degradation [4].

AGO super-family members present strong evolutionary conservation, are ubiquitously expressed, and can be sub-divided into AGO, P-element-induced wimpy testis (PIWI), and SAGO sub-families [5]. The AGO subfamily includes four different proteins in humans: AGO1, AGO2, AGO3, and AGO4 with extremely high homology (exceeds the 80% value over the entire protein-length) among all members (Figure 1). They share the same signature domains N, MID, PAZ, and PIWI with PAZ and PIWI being identified in proteins implicated in RNAi-mediated processes [6,7,8]. Thereby, it seems that the molecular discrimination between members of the AGO subfamily represents a complicated and tangled issue difficult to be resolved. It has been previously suggested that subfamily-member functions can be overlapping and/or mutually compensatory when required [9,10,11]. Although, in humans, only AGO2 has proved to exert slicer-endonuclease and microRNA (miRNA) stabilization activities [12,13,14,15], which is a selective cleavage activity for AGO3 that has been recently discovered [16]. AGO3 has been engaged in mRNA decay via Alu-elements mediation directed by RNA [17]. AGO1 executes a more subsidiary role in gene silencing, whereas AGO4 seems to control the entry into meiosis and sex-chromosome silencing in the mouse germ line [18,19,20]. The leading canonical pathway of miRNA biogenesis, achieved by AGOs, commences with transcription of the pri-miRNAs, which are being processed into pre-miRNAs by the micro-processor complex, the DiGeorge Syndrome Critical Region 8 RNA binding protein (DGCR8), and Drosha, which is a ribonuclease III enzyme [21]. Then, Exportin 5 (XPO5)/RanGTP complex undertakes the export of pre-miRNAs to the cytoplasmic compartment, where they are cleaved by Dicer, which is an RNase III endonuclease [21,22]. This pace involves the withdrawal of the terminal loop, which, ultimately, leads to miRNA maturation [23]. Hereafter, the mature miRNAs are able to be loaded onto AGO complexes in an ATP-dependent manner [24]. The passenger strand is ultimately released while the preserved strand guides and leads to the advancement of gene expression regulation. The retained miRNA guide strand facilitates the transfer of AGO-miRNA complex to the 3′-UTR (untranslated region) of mRNA targets. A crucial aspect is the degree of sequence complementarity between the miRNA seed region (2–8 nucleotides from the 5′-end) and the mRNA target. The miRNA:mRNA matching ensues translational repression and destabilization, and/or degradation of the bona fide mRNA target sequences [25].

In recent years, besides the canonical pathway, a number of non-canonical biogenesis pathways have emerged [39,40]. In non-canonical biogenesis, the cleavage of microprocessor complexes is avoided, or alternative combinations of Exportin and/or miRISC loading complex proteins are employed [41,42].

The versatile functions of AGO can be tightly regulated by several post-translational modifications such as SUMOylation [43], acetylation [44], and ubiquitination [45,46]. Moreover, under cell stress, AGOs are modified by poly (ADP-ribose) in order to relieve miRNA-guided repression [47,48]. The phosphorylation state of a critical serine/threonine cluster is essential for AGO-miRNA interaction(s), which affect mRNA binding and localization. Especially, phosphorylated Ser^387^, Tyr^393^, and Tyr^529^ have proved important for miRNA-guided gene silencing in vivo [49,50]. Remarkably, mutation of Pro^700^ to Ala^700^ leads to AGO2 destabilization, which couples Pro^700^ hydroxylation of AGO2 with regulation of its stability. In an attempt to measure the efficiency of AGO hydroxylation in vitro, Qi and colleagues showed that AGO2 and AGO4 are more efficiently hydroxylated than AGO1 and AGO3 [51], which indicates differential activities and regulation patterns of AGO subfamily members.

Through the constantly accumulated information with regard to the modifications of AGOs and their subsequent functional alterations, their fundamental engagement in multiple human pathologies has recently emerged. Given the existing and extensive literature describing the miRNA profiles of specific diseases, this study will entirely focus on analyzing the pathogenic effect(s) of AGO alterations specifically, in terms of their expression, polymorphisms, modifications, and interactions. This review aims to uncover the consequence of AGO deregulation(s), as causal effectors, in viral infections, autoimmune diseases, tumorigenesis, progression, metabolic deficiencies, mental disorders, neuronal diseases, and infertility.

## 2. AGOs in Human Diseases

### 2.1. AGOs in Viral Infections

The precise role of RISC/miRNA machinery in the host-pathogen interactions is still unclear. In this section, we discuss recent findings that highlight the importance of AGO in viral infections, and their impact on viral replication and immune response. Comprehensive understanding of AGO biology will aid the illumination of molecular mechanisms underlying human viral infections and the reinforcement of prompt development for successful miRNA-based therapies.

Viral miRNAs were initially reported to be expressed by the human γ-Herpes Virus Epstein-Barr Virus (EBV) [52], which was followed by the subfamily members of Herpes Viruses, including the Marek’s Disease Virus (MDV) [53], the (α-Herpes Viruses) Herpes Simplex Virus 1 (HSV-1) [54], the HSV-2 [55], and the (β-Herpes Virus) human CytoMegaloVirus (hCMV) [56]. miRNA expression changes were also observed in members of a variety of DNA-virus families, such as the PolyomaVirus, but were absent from other DNA-virus families, such as the papilloma viruses, pox viruses [57], and retro viruses [52]. These miRNAs may hold strong potential in RNA-silencing activities in viruses and/or host-virus interactions. Ιn mammals, the viral infection and spreading are affected by innate-immune responses related to Interferon (IFN)-induced signaling pathway(s) [58,59,60,61,62,63]. Given that a double-strand RNA (dsRNA) sequence longer than 30 bp can efficiently trigger IFN-dependent pathway(s) activation [61,64], it is reasonable to suggest that the naturally occurring mechanism of RNAi has been evolutionarily compromised or inactivated [65,66].

The mechanistic implication of small RNAs in viral infections was thoroughly investigated in invertebrates. In these organisms, the inhibition of viral replication is achieved by small-RNA production via the viral dsRNA template, which triggers viral-RNA degradation [67,68]. Responsible molecules for the precise function of the innate-immune system are the Pattern Recognition Receptors (PRRs) [69] that mediate the release and dispensation of inflammatory cytokines [70,71] by recognizing the virus-conserved pathogen-associated molecular patterns (PAMPs) [67,72]. A discrete family of PRRs is the dsRNA-specific dicer nucleases. The initiated trigger for dicer-mediated immunity against RNA and DNA viruses is related to specific PRRs. Dicer cleaves the viral-RNA trigger to generate virus-derived small RNAs (viRNAs), which provides antiviral defense through RNAi-silencing activation [73,74]. To counterbalance this host reaction, viruses produce intrinsic virulence factors called viral suppressors of RNA silencing (VSRs) [73]. These molecules presume upon host mechanisms to thrive in hostile micro-environments [75], which provides protection from Dicer cleavage and/or impeding viRNAs to be loaded onto AGO proteins [76,77,78]. RNAi-machinery components interact with VSRs, which suppresses the efficiency of viral-RNAs cleavage [79,80,81] and prohibits AGO from catalysis of the RNA-cleavage reaction [60] (Figure 2).

Remarkable interactions of RISC-complex components with viral nucleic acids have been previously reported across the heterogeneous virus families. In mammals infected by Hepatitis B Virus (HBV), miR-20a-loaded AGO2 translocation to the nucleus strongly suggests that the miRNA-directed Transcriptional Gene Silencing (TGS) mainly depends on this binding partnership. This leads to methylation of HBV DNA, which, next, causes suppression of HBV replication [85]. Upon HIV infection, strong reduction of viral production is observed in response to AGO inhibition [86]. In HCV infection, miR-122, an abundant miRNA species in liver cells, seems to facilitate viral replication without affecting mRNA translation or RNA stability [87].

A serious controversy has prevailed in the field concerning the antiviral-defense repertoire in mammals due to lack of a typical (canonical) mechanism that relies on the siRNA-dependent AGO-group functionality [88,89,90,91]. This is further reinforced by pioneer observations being related to the copy-number of miRNAs that is able to critically influence host-transcripts and/or viral-transcripts homeostasis [92]. Strikingly, previous seminal reports have shown that deficiencies of Dicer and AGO functions in pluripotent mouse Embryonic Stem Cells (mESCs) can be well tolerated [9,93]. In such settings, the RNAi response by a long-dsRNA trigger presents dominant action and constitutes a major layer of defense in the absence of the IFN response [94,95]. Deep-sequencing analysis unveiled the accumulation of a number of miRNA species in undifferentiated mouse cells being infected with encephalomyocarditis virus (EMCV) and nodamura virus (NoV) [60]. KLF4, Oct4, and Sox2, which are bona fide reprogramming factors that induce cell pluripotency, are presented to operate incompatibly with IRF7, which is a key regulator of IFN-mediated antiviral defense. Most likely, the rapid cell division of pluripotent stem cells may be functionally contradictory to IFN-associated responses controlling anti-proliferative activities [96].

The RNAi-response program is well-conserved across animal species and has not been replaced by the IFN system [95,97]. The identification of siRNAs, for many years, was ambiguous due to detection of potent RNAi suppressors and high IFN levels, especially in somatic differentiated cells. Restoration of the production of viral siRNAs was observed in murine somatic cells carrying *3A*-deficient mutants of human EntEro-Virus 71 (HEV71), and demonstrating the role of the 3A protein as an RNAi suppressor [98]. Furthermore, loading of these siRNAs onto AGOs provokes the cognate viral-RNA genome’s degradation. Pharmacological blocking of the IFN response and utilization of mouse cells with lack of functional Type I IFN-receptor argue that the RNAi response does not represent an IFN-mediated effect.

In 2017, Wang and colleagues discovered that H5N1 viral infection prompts the reduction of AGO2 nuclear distribution in lung carcinoma cell line. AGO2 was shown to negatively regulate the Type I IFN-signaling pathway by competitive binding to IRF3 (a major effector of the IFN-β pathway) with the CBP/p300 transcriptional co-activator. Upon viral infection, the nuclear AGO2 content could be reduced, which provides inhibitory signals for AGO activity and stimulatory cues for IFN-β expression, and signaling. Thus, consequently, this promotes viral propagation [99]. This was not unforeseen since Dicer affects influenza virus replication in an siRNA/miRNA-independent manner [100]. The same cell line was used to examine the transcription levels of the RNAi-machinery components Drosha, DGCR8, and Dicer in the presence of Dengue virus (DENV). Knock-down of these genes caused increased viral replication rates [101].

A number of studies in recent years have reported the differential expression and function of miRNAs in airway cells [102,103], which significantly affects processes such as modulation of innate and adaptive immune responses [104] due to viral respiratory infections. Such viruses include the orthomyxovirus influenza and the severe acute respiratory syndrome Corona virus 2 (SARS-CoV2). Since the availability of effective vaccines and antiviral treatments against respiratory viral infections are limited [105,106], novel antiviral approaches are necessitated to be promptly developed. Promising results rise from experiments engaging the mouse hepatitis virus (MHV), which is a Corona virus that is closely related to the SARS-CoV2. In mammalian cells, the replication of MHV increased when the Nuclei capsid (N) protein was expressed suppressing RNAi triggered by either short-hairpin (sh) RNAs or small-interfering (si) RNAs. MHV replication was also enhanced by knocking-down the expression of Dicer and AGO2 [107]. These results provide evidence for new key contributors in the antiviral immunity responses, such as AGO, in mammalian cells.

A recent study by Jeffrey and colleagues [108] described the antiviral role of AGO4 in mammalian cells for the first time. They demonstrated that AGO4 has an IFN-boosting property and an antiviral RNAi activity. This function was shown to be independent of the other AGOs, IFN activity, and RISC loading. In their experiments, they reported that immune cells with AGO4 deficiency and AGO4 knockout mice were significantly more susceptible to viral infections. It remains to be determined whether this function is virus-specific and how AGO4 expression could be increased to control viral infections. Furthermore, studies of influenza-infected mice have unearthed that *AGO4*-deficient animals carry significantly higher viral loads. It is important to determine the spectrum of viral types whose infection capacities can be affected by AGO4-expression levels. Better understanding of how the immune system functions, and how AGOs are mechanistically implicated in its homeostasis, integrity, and plasticity will allow the generation of efficient and systemically safe defense treatments against a wide range of pathogenic viruses.

### 2.2. AGOs and Autoimmune Diseases

AGOs are critically implicated in several auto immune diseases (AIDs) such as auto immune thyroid diseases (AITDs), autoimmune encephalomyelitis (EAE), Crohn’s Disease (CD), idiopathic sudden sensorineural hearing loss (ISSNH) and rheumatic diseases (RDs). AGOs are genetically linked to AITDs, such as the Graves’ Disease (GD) and Hashimoto’s Disease (HD), through gene polymorphisms observed in *AGO1* and *AGO2* DNA sequences. More specifically, the *AGO1rs636832* and *AGO2rs11166985* polymorphisms were more frequently observed in GD patients than in healthy individuals (controls) while the *AGO2rs11166985* and *AGO2rs2292779* were more commonly detected in clinically intractable GD cases. Elevated expression of *AGO1* mRNA was observed in AITD patients while *AGO2* mRNA contents were increased in intractable GD patients than in individuals with GD in remission [109]. In ISSNH, a medical disorder with unknown aetiology and pathogenesis [110], AGO2 was found upregulated in the peripheral blood of patients, which strongly suggests the major contribution of AGO proteins to a wide spectrum of AIDs [111]. Moreover, deregulated levels of AGO2 were reported in CD, which is a chronic idiopathic inflammatory bowel disease. The microbial disruption of autophagy leads to expression changes of AGO2 and to a subsequent abnormal miRNA expression that drives and promotes CD pathogenesis [112].

The role of miRNAs in AIDs, including EAE, is under extensive investigation [113]. Global regulation of miRNA expression in the brain of immunized mice with EAE pathology was shown to depend on the availability of AGO2. miRNAs were loaded onto AGO complexes, and their availability and interactions with other RISC proteins may play important roles in the complexity of post-transcriptional regulation of the immune response [114,115].

Auto-antibodies are an important aspect of AIDs. Anti-Su auto-antibodies have been previously reported to be preferentially reactive against native antigens, as revealed by the Double Immune-Diffusion (DID)-protocol employment [116]. Twenty years of thorough research were required for the characterization and identification of AGO2 as the critical target antigen [117,118]. Anti-Su/AGO2 autoantibodies do not seem to bear any major disease specificity since they are detected in 10%–20% of patients with different RDs including the Systemic Lupus Erythematosus (SLE), Polymyositis (PM), Dermatomyositis (DM), Scleroderma (SD), and Sjögren’s Syndrome (SS), and even in apparently healthy individuals at lower prevalence [119]. However, other studies reported that Anti-Su auto-antibodies prevalence was 3% in SLE, 5.6% in probable SLE patients, 0% in Rheumatoid Arthritis (RA) and PM, and 3.3% in SD and Systemic Sclerosis (SSc) [116]. Anti-Su positive SLE patients compared to other published series of SLE cases were presented with an increased prevalence of Raynaud’s Disease (RD), and a reduced prevalence of malar rash, alopecia, arthritis, and anemia [120]. Further research needs to be performed to clarify if there is an actual anti-Su specificity regarding the diverse RD clinical and molecular pathologies.

Anti-Su identification is not limited to SLE patients and other common systemic AIDs [121], but is expanded in rare conditions such as the primary anti-phospholipid antibody syndrome (PAPS). A 13% ratio of PAPS patients were found to carry anti-Su and 10% were found positive for the anti-Ro60, which are similar to the anti-Su ones [122] in contrast to the absence of other auto-antibodies [123]. Anti-Su was also observed in the undifferentiated connective tissue disease (UCTD) [116,124], which dictates anti-Su for being a common auto-antibody in atypical RD cases. Their mechanistic role and clinical significance remain to be elucidated in order to provide novel and powerful tools for an early and accurate diagnosis, and, in some patients, to observe the progress of RD pathology.

Notably, although the anti-Su counteracts with AGO2, anti-Su-positive sera can also react to AGO1, AGO3, and AGO4 protein antigens likely due to the high conservation among their respective sequences [118]. Interestingly, 5% of Hepatitis C Virus (HCV) or Hepatitis B Virus (HBV) + HCV co-infected but not in HBV patients were anti-Su/AGO2 positive [125]. However, IFN treatment did not lead to a detectable effect in autoantibody production despite its major implication in autoimmunity repertoires [120]. Identification of the components of RNAi machinery, such as the AGO proteins, as targets of the anti-Su/AGO2 autoantibody system, implements an autoimmune response directed at the macromolecular complex and implicated in post-transcriptional regulatory scenarios of gene expression. Further investigation is definitely required to undoubtedly unveil AGOs as master players in the development and progression of AID pathologies.

### 2.3. AGOs in Cancer

The essential engagement of AGO in cancer has been extensively studied, and many research reports and review articles have identified the miRNA populations whose expression is altered during tumorigenesis. Thereby, this review has focused on AGO with emphasis on their expression, polymorphisms, mutational profiles, modifications, and interactions in cancer.

The deregulated *AGO*-gene expression is observed in a variety of malignancies such as breast cancer, melanoma, ovarian, urothelial, prostate, clear-cell renal, gastric, and colorectal carcinomas, as it seems to affect the proliferation rate and/or migratory potential of tumor cells [126,127,128,129,130,131,132]. Elevated expression of AGO is often reported in glioma, breast, hepatocellular, gastric, colon, ovarian, bladder, and prostate cancers, even though there are incidents of lower levels detected, as graphically illustrated in Figure 3. Expression studies for breast, lung, prostate, gastric, and renal cancers as well as glioma, melanoma, and acute lymphoblastic leukemia support the heterogeneity of AGO-expression levels in cancer. A detailed annotation regarding the changes (increase or decrease) of AGO expression across different cancer types is presented in Table 1. The deregulation of AGO2, which is the most abundant member of the family, has been associated with tumorigenesis and cancer progression and, therefore, it has been suggested for therapeutic intervention and treatment [133,134,135,136]. Using GeneMANIA (Multiple Association Network Integration Algorithm) [137] for prediction of gene function, Liu and colleagues revealed that AGO2 upregulation was correlated with acceleration in tumor progression and poor survival in a cohort of 962 lung-cancer patients [138]. However, in melanoma cells, downregulation of AGO2 is able to confer cell and tumor-growth [139]. Since AGOs exhibit cell-type-dependent and tissue-dependent sub-cellular-distribution patterns, and different regulation profiles of gene expression in between normal and oncogenic cellular settings [140], opposite functionalities are expected to be deployed by following tumor-specific fashions.

The *AGO2* chromosomal position, 8q24.3, constitutes a frequently amplified locus in many cancer types, including the Hepatocellular Carcinoma (HCC). The change of DNA-copy number of AGO2 through DNA amplification or gene-gain processes may lead to upregulation of AGO2 cellular activity. This can, subsequently, affect the transcription levels of Focal Adhesion Kinase (FAK) in HCC patients [156]. FAK is a Non-Receptor Tyrosine Kinase, activated by Integrins and Growth Factors that can influence the cytoskeleton structures, cell-adhesion sites, and membrane protrusions, and, therefore, control cell movement in cell-migration and angiogenesis processes [171]. Elevation of FAK occurs by miRNA-independent AGO2 function. In addition, the AGO2 slicer-independent role in the miRNA-stabilization process [156,172] enhances the microRNA levels post-transcriptionally, which offers a new potential target for promising HCC treatment and therapy.

Single-Nucleotide Polymorphisms (SNPs) of *AGO1* and *AGO2* genes were found to be related to Disease-Free Survival (DFS) and Overall Survival (OS) in breast cancer in order to risk impact of renal cell carcinoma [173,174,175,176]. More specifically, *AGO2 rs3864659* presents a protective effect on breast cancer patients whereas *AGO2 rs11786030* and *rs2292779* have been linked to a poor prognosis in Korean cohorts [177]. Furthermore, the *AGO1 rs595055* polymorphism has been associated with low breast cancer risk in Russians [178]. In a Chinese-Han population, a genetic variant (AA and A allele of *rs636832*) of *AGO1* influences gastric cancer by imposing a lower level of lymphatic metastasis [179]. In a Caucasian male cohort, the AG + GG genotypes of *AGO1*
*rs595961* were shown to carry a significant protective effect in renal cell carcinoma [180]. Additionally, the same polymorphism seems to be associated with lung cancer risk in a Chinese female case report [181]. The impact of genetic variants *rs636832* and *rs595961* located within the *AGO1* genetic locus may deliver susceptibility for specific types of cancer such as acute and chronic lymphoblastic leukemia, lung, renal, and bladder cancer [182].

Besides the deregulation of AGO-expression levels and SNP-dependent oncogenicity, post-translational modifications of AGOs can lead to an increase of the invasive potential of tumor cells. Under normoxia, c-Src kinase appears to obtain a stronger binding affinity for AGO2 and, therefore, it presumably constitutes the major regulatory kinase for AGO2. Nevertheless, phosphorylation at the critical Tyr^393^, Tyr^529^, or/and Tyr^749^ residues of AGO2 protein compels a tumor-promoting response [138]. Clinical data from breast cancer patients unveil a link between AGO2 phosphorylation at Tyr^393^ and poor overall survival [183]. This phosphorylation, dictated by EGFR but not c-Src kinase, inhibits Dicer binding to AGO2 and, subsequently, maturation of a particular miRNA subset. Notably, hypoxia-mediated prolyl-4-hydroxylation at Pro^700^, which is another crucial modification, causes accumulation of AGO2 protein in stress granules by associating with the Hsp90 molecular chaperone. This dictates the elevation of miRNA levels, which influences protein stability and affects tumorigenesis and tumor growth [184]. Besides AGO2, AGO1 serves as a target of hypoxia since in silico bioinformatics algorithms and in vitro validation assays proved that the Hypoxia-Induced Factor 1α (HIF1α) promotes Hypoxia-Responsive miRNAs (HRMs) that target AGO1 [185]. Subsequently, AGO1 induces translational de-suppression of *VEGF* mRNA associated with tumor angiogenesis and poor prognosis, and, thereby, provides a therapeutically new, beneficial target for an anti-angiogenesis or pro-angiogenesis strategy.

DNA damage is an important contributor to tumor formation as Double-Strand Brakes (DSBs) are hot spots of genome instability with demanding need of proper, efficient, and prompt repair. DNA-Damage Response (DDR) foci formation studies and checkpoint assays in human, mouse, and zebrafish demonstrated that elements of the RNAi pathway, such as Dicer and Drosha, but not AGOs, are necessary to activate DDR upon exogenous-DNA damage and oncogene-induced genotoxic stress [186]. Wei and colleagues, in a seminal article, reported that double-strand break-induced small RNAs (diRNAs) are produced in close vicinity of DSBs, which, then, serve as guiding molecules to facilitate the DNA-repair process. AGO2/diRNAs can either directly recruit DSB repair proteins or assist in modifying local chromatin and indirectly facilitating the repairing course [187]. How exactly AGO proteins are involved in DSB repair is a fascinating question that remains to be addressed. Gao and colleagues showed that AGO2 forms a complex with Rad51, which is a nucleoprotein-filament. The complex is able to facilitate strand invasion and to initiate the Homologous Recombination (HR) process [188]. Rad51-protein accumulation at DSB sites and HR repair depends on the catalytic activity and small RNA-binding capacity of AGO2. These findings strongly support a model in which Rad51 is guided to DSB sites by diRNAs via AGO2-specific interaction [189].

Lastly, AGOs seem to play a regulatory role in chromosomal-telomere integrity that has proved indispensable for maintaining genome’s stability and controlling cell proliferation. The length of human telomeric DNA is maintained by telomerase, which is a ribonucleoprotein (RNP) enzyme whose activity is highly elevated in 85%–90% of human cancers [190,191]. Telomerase contains two catalytic components: the Telomerase Reverse Transcriptase (TERT) and the H/ACA box Telomerase RNA Component (TERC), which is the template used for the synthesis of telomeres. AGO2 promotes telomerase activity, and stimulates the association between TERT and TERC, by interacting with TERC and interacting with a newly identified sRNA (terc-sRNA). In accordance, AGO2 depletion results in shorter-length telomeres as well as in lower proliferation rates, both in vitro and in vivo [192]. Altogether, these data uncover a new layer of complexity in the regulation of telomerase activity by AGO2, and may lay the foundation for novel therapeutic targets in tumors and telomere pathologies.

### 2.4. AGOs in Metabolic Deficiencies

#### 2.4.1. Mitochondrial Dysfunctions

Accumulating evidence demonstrates the extensive association of AGOs with mitochondrial function. It has been suggested that AGO2 acts as a key mitochondrial translation-initiation factor to facilitate ribosome-mRNA interactions [193]. A set of 13 miRNAs, referred to as mitomiRs, was significantly enriched in mitochondria of HeLa cells with 120 target sites along the mitochondrial DNA (mtDNA) sequence. The most frequent predicted target sites were located at transcripts of ND1, ND4, ND5, and ND6 proteins, which are all components of the first complex of the respiratory chain, as well as the COX1 and COX2 ones, which are responsible for formation of prostanoids, including prostaglandins [193]. Proteomic studies have identified mitochondrial proteins mostly from the inner membrane to act as AGO2-binding partners, including a number of ATP/ADP translocases, carriers, and ribosomal proteins [194].

In pathological conditions, such as diabetic insult, the cardiac mitomiR distribution patterns are dramatically altered. This is not surprising, as type 1 diabetes mellitus (DM) is related to cardiac functional deficits, which may be derived from a diminished ability of cardiac mitochondria to generate ATP. Jagannathan and colleagues proved that, when AGO2 content is restricted, and the mitomiR-378, is absent, the mitochondrial RISC-mediated regulation is abolished. A balanced mitochondrial-RISC component stoichiometry assists in the formation of a functional complex and dictates the loss of mitochondrial-encoded proteins in the diabetic heart [195].

The significance of communication between mitochondria and nucleus has recently emerged. Despite the fact that published information shows the existence of miRNA machinery inside mitochondria [193,196,197,198,199,200,201], little is known concerning the AGO-mediated crosstalk in between the mitochondria and nucleus. AGO2 may be selectively imported into mitochondria through a target sequence predicted to reside near the NH_2_-terminus of the AGO2 protein [193]. This observation, together with the recent finding that AGO2 associates with Hsp90 [202], can provide an interesting rational of a molecular mechanism for the transfer and entrance of AGO2 into mitochondria as well as its pivotal role in the organelle.

Apart from miRNAs, a new class of small non-coding RNAs (sncRNAs), the mitochondrial tRNA Fragments (mt tRFs), was identified as a contributor to mitochondria-nucleus communication. In the clinical phenotype of oxidative phosphorylation (OXPHOS) diseases, a Dicer-dependent and AGO2-dependent accumulation of selected mt tRFs was reported, which suggests that mt tRFs are subjected to a mechanism of action similar to the miRNAs recognized one. Intriguingly, by using the cybrid model of MELAS (Mitochondrial Encephalomyopathy, Lactic Acidosis, and Stroke-like episodes), a causal mutation (m.3243A > G) in the mitochondrial *tRNA^Leu^*^*(UUR)*^ gene was identified. This gene could change the production of mt tRFs [203].

Two mechanisms of mt tRF biogenesis and function can be proposed, depending on Dicer and AGO2 localization. (I) mt tRNA molecules exported out of mitochondria are cleaved by Dicer, providing mature mt tRFs. Cytosolic mt tRFs are loaded onto AGO2 for silencing nuclear-encoded genes, such as the *Mitochondrial Pyruvate Carrier 1* (*MPC1*). (II) The other proposed mechanism suggests the cleavage of mt tRFs followed by loading of their mature versions onto mtAGO2 proteins critically participating in regulation of the expression of mtDNA-encoded genes [201,203]. Regardless of the sub-routine via which they are produced, tRFs can associate with AGO proteins to form biologically active complexes [204,205]. The 3′ tRFs are more effectively associated with the AGO3 and AGO4 than the AGO1 and AGO2 subfamily members. It seems that tRFs are able to negatively regulate gene expression, and, although few endogenous targets for tRFs have been identified so far [205,206,207,208], their precise mode of production and function remain elusive [209]. Further related studies will greatly advance our knowledge regarding mitochondrial malfunction and their implication in pathological conditions of human diseases.

#### 2.4.2. Obesity

Obesity is a worldwide, serious health problem associated with an increased risk of life-threatening diseases such as diabetes, fatty liver diseases, atherosclerosis, and certain types of cancer. The energy balance (production versus consumption) is a crucial aspect in these conditions, in which AGOs, and in particular AGO2, can be actively engaged as a key component of glucose and lipid metabolism. The AGO-mediated regulation of pathology being manifested in chronic metabolic disorders and obesity-associated sequelae can be conducted by miRNA-dependent and miRNA-independent processes, and can provide beneficial, new information regarding the pathogenesis and treatment of obesity.

It is well established that obesity causes defects in autophagy in the liver, leading to poor mitochondrial quality control [80,210,211] and to impaired protein synthesis, which induces an energy sourced-accumulation. The activation of AMP-activated protein kinase (AMPK), which is a kinase that serves as a major cellular-energy sensor, and its substrates ULK1, MFF, and PGC1α, improves mitochondrial capacity and quality, while, subsequently, generates sufficient energy for protein synthesis [212]. These processes can be controlled by an miRNA-independent mechanism through the AGO2-CCR4-NOT and AGO2-PAN2-PAN3 exonuclease complexes [25] that cooperate to likely repress protein translation and energy metabolism in the obese liver.

Hepatic AGO2 can also regulate the energy expenditure during the course of obesity by RNA silencing through the expression of a subset of miRNAs, including miR-802, miR-103/107, and miR-148a that target the *HNF1β*, *CAV1*, and *AMPKA1* genes, whose products are critically implicated in glucose and lipid metabolism and homeostasis. Importantly, *AGO2* inactivation protects from obesity-associated glucose intolerance and hepatic steatosis in mice [212]. The significant upregulation of Drosha, DGCR8, Dicer, and AGO2 in human umbilical vein endothelial cells (HUVEC) under hyperglycemic conditions provides strong indications for the essential contribution of AGO2 and its interactors to metabolism pathologies [213].

A fundamental, but still unanswered, question is how individual cells manage their intracellular metabolism with a simultaneous maintenance of the ongoing cellular functions to avoid hyperglycemic stress. In pancreatic β-cells, loss of *AGO2* and *miR-375* abolished compensatory proliferation of cells in response to an increase in the demand for insulin during insulin resistance and hyperglycemia [214,215]. The increasing demand for insulin clearly guides to an miRNA-mediated balance between the energy requirements organized for proliferation versus those for exocytosis in the pancreas [214]. In pancreatic β-cells, loss of *AGO2* abolishes compensatory proliferation during insulin resistance, which indicates a potential role in an adaptive capacity [50,216]. AGO2 expression, in these cells, is modulated in accordance to glucose metabolism, which provides further evidence for its direct link to cellular-energy homeostasis [217]. Moreover, in liver, AGO2 expression is modulated according to changes in extracellular glucose concentrations. Conditional deletion of *AGO2* in hepatocytes diminishes adaptive-glucose production during fasting, while its loss promotes the expression of key-metabolic molecules, such as AMPKa1, by derepressing miR-148a. In hyperglycemic, obese, and insulin-resistant Lep^ob/ob^ mice, *AGO2* deletion reduces random and fasted blood-glucose levels and body weight, and improves insulin sensitivity [214]. Altogether, these results identify AGO2 as an essential regulator of the fasting response in liver and further reinforce the critical role(s) of AGOs in adaptive-physiological processes.

### 2.5. AGOs in Psychiatric Disorders and Neuronal Diseases

RISC machinery is an important regulator of gene expression in stress-related psychiatric pathologies. Molecular analysis and behavioral studies in mice indicate that AGO2-associated miRNAs and cognate mRNA targets play an important role in these conditions, as the AGO-miRNA complex is a key regulator of intact behavioral response(s) to chronic stress (Figure 4). Chronic stress leads to increased miR-15a levels in the amygdala-AGO2 complex and a simultaneous reduction of its predicted target, *FKBP51,* whose protein product is involved in stress-related psychiatric disorders [218]. Implication of the RISC machinery was reported in post-traumatic stress disorder (PTSD) of war veterans (WV), validated by the clinician administered PTSD scale (CAPS) [219], and the Diagnostic and Statistical Manual of mental disorder (DSM-V) [220]. In particular, there is deregulation of the miRNA biogenesis pathway due to diminished expression of its key molecules *AGO2*, *DCR1,* and *STAT3* [221]. Even in the case of cocaine addiction, AGO2 has been significantly involved. In such a situation, elevated levels of released dopamine are followed by stable changes in gene transcription, mRNA translation, and metabolism, within medium spiny neurons in the striatum. A distinct group of miRNAs was identified that could be specifically regulated by AGO2 in the striatum and was suggested to likely play a role in cocaine addiction. Furthermore, *AGO2* deficiency in Dopamine 2 Receptor (Drd2)-expressing neurons greatly reduces the motivation to self-administered cocaine in mice [222]. The engagement of AGO2 in mental conditions is reinforced by its critical implication in Fragile X Syndrome (FXS), which is a disease of aberrant protein production and behavioral abnormalities that include autistic-like features [223,224,225,226]. The Fragile X Mental Retardation Protein (FMRP) is required for normal cognition and, as a result, FXS patients are cognitively impaired. AGO2 activity is facilitated or blocked by MOV10, which is an RNA helicase that participates in modulating miRNA targeting [227]. MOV10 binding was shown to be located on accessible region(s) at the beginning of 3′-UTR, ~21 nucleotides residing downstream of the terminating ribosome [228]. MOV10 allows the AGO2 binding to formerly inaccessible miRNA Recognition Elements (MREs), by unwinding of GC-rich secondary structures. When FMRP recognizes and interacts with 3′-UTR, AGO2 binding is blocked, which likely uncovers a competitive association of MOV10 for the same GC-rich sequence [229]. Hence, the MOV10 function as helicase may be prevented by the FMRP ability to stabilize the structure with subsequent impact on AGO2 accessibility to MREs in the brain [229].

Furthermore, *AGO1* genomic alterations present a causal link to intellectual disability/autism spectrum disorder. A critical *AGO1*-gene mutation (Gly199Ser) is associated with recognizable clinical features such as hypotonia, infrequent seizures, and intellectual disability, and facial features consisting of telecanthus, a wide nasal bridge with a bulbous nasal tip, and a round face with downwardly slanted palpebral fissures. However, telecanthus closely resembles the phenotype obtained from patients carrying a chromosomal micro-deletion encompassing the *AGO1* locus [230]. Thereby, *AGO1* mutations may serve as causality for the syndromic form of intellectual disability/autism-spectrum disorder.

AGOs are also engaged in neurological diseases, such as Huntington’s and amyotrophic lateral sclerosis (ALS). In mouse models, as well as in post-mortem material from patients with Huntington’s Disease (HD), AGO2 accumulation is observed when neuronal protein aggregates are formed, as a result of impaired autophagy. AGO2 accumulation is essentially different in between mature non-dividing neurons and dividing cells, and results in global changes of miRNA activity, which contributes to alterations of post-transcriptional networks in neurodegenerative diseases [231].

In the case of ALS, a fatal progressive, neurodegenerative disease that targets motor neurons (MNs), the mitochondrial dysfunction is one of the most crucial molecular pathologies developed [232]. The mechanistic coupling between mitochondria and RNAi machinery has emerged as an exciting and powerful area, beginning with the discovery that AGO2 associates with mitochondria [233]. Specifically, AGO2 was found selectively bound to mitochondrial tRNA^Met^, which provides evidence that links mitochondria with RNAi-related components, such as AGOs. Notably, AGO2 and Dicer were detected in isolated mitochondria derived from hippocampal neurons [234,235]. Further data suggest that mitochondria participate in the axonal localization and transport of RNAi machinery, and alterations in this mechanism may be associated with neurodegeneration in ALS [236].

Recent studies have also directly connected ALS pathology with deregulation in global [237,238] and local protein synthesis [239]. Strikingly, AGO2, Dicer, and miRNAs were shown to reside in distal axons and growth cones. The large number of mRNA transcripts discovered in both dendrites and axons suggests that local protein translation is the rule and not the exception [240,241,242]. Although it is not fully understood how mRNA transcripts can be localized at distinct neuronal compartments in MNs, the majority of mRNAs is translationally repressed in the axon(s), waiting for an extrinsic cue (i.e., stress, trophic factor, or injury) to be applied. This repression is crucial during axonal transport and recent work has suggested that loaded miRNA machinery critically contributes to this silencing course [243]. The regulation of local protein synthesis in growth cones and axonal junctions by anchoring RISC proteins and miRNAs, a process disrupted in mutant SOD1 (SuperOxide Dismutase 1) neurons of ALS pathology, plays an important role in the alterations of local biosynthesis [236]. Altered RNA localization and local-protein synthesis can be toxic to neurons, and have been related to ALS and other neurodegenerative diseases, such as spinal muscular atrophy (SMA), HD, spinocerebellar ataxias (SCAs), Parkinson’s disease (PD), Alzheimer’s disease (AD), and prion diseases [241,244,245,246]. Taken together, it seems that the contribution of AGOs to mental and neurological diseases is decisive, but requires further assessment to thoroughly evaluate its functional causality for neuropathology development.

### 2.6. AGOs and Infertility

Human infertility affects many couples with almost 7% of healthy men suffering from involuntary childlessness for which clinical examinations do not provide answers [247]. Extensive research has been performed, which aimed to unravel causalities of different types of infertility. Evidence has been provided for the direct and indirect implication of AGOs both in spermatogenesis and oogenesis.

The successful completion of gametogenesis requires a comprehensive programming of gene expression. In *C. elegans*, ALG-3 and ALG-4 AGO proteins were found to promote male fertility [248,249]. The other AGO-family member, CSR-1, which interacts with 22G RNAs (~22-nucleotide-long small RNAs, predominantly having a 5′-G and being synthesized by RNA-dependent RNA polymerases, using long transcripts as templates) to protect the mRNAs from Piwi-piRNA mediated silencing has been identified in the sperm, together with small anti-sense RNAs against oocyte-specific germ-line mRNAs. The other Argonaute family member CSR-1, which interacts with 22G small RNAs to protect the mRNAs from Piwi-piRNA mediated silencing, is also present together with small RNAs anti-sense to female-specific germline mRNAs into the sperm. Functions of ALG-3/4 and CSR-1 are prerequisites to promote robust spermatogenic-gene expression. Although ALG-3/4 are absent from mature sperm [248], CSR-1 contents have proved to be abundant in the same condition. These two seem to act in synergy, providing an epigenetic memory of paternal gene expression via sperm. In accordance, heterozygous offspring produced from homozygous *alg-3/4* or *csr-1* males crossed with a wild-type exhibit reduced fertility [250].

In mammals, nuclear localization of AGO4 in germ cells during prophase I of meiosis and its accumulation within the sex body of pachytene spermatocytes indicate AGO4 implication in the silencing process of meiotic sex chromosome inactivation. Likewise, an RNA-induced transcriptional silencing complex (such as in *S. pombe* and *C. elegans* [251]) exists and operates at the heterochromatin level during meiosis [19,251]. Importantly, ~20% of the global miRNA downregulation is derived from miRNAs expressed from the X chromosome, and, strikingly, all miRNAs encoded on the X chromosome are significantly less abundant in AGO4^−/−^ mutant cells [19]. Meiotic silencing includes autosomal regions that fail to pair with their homologous partners, where AGO4 is located [252]. AGO4^−/−^ male mice are shown to be sub-fertile, with reduced testis size and lower epididymal-sperm counts. Lack of AGO4 causes impairment of meiotic prophase I and, consequently, high proportion of apoptotic-prone spermatocytes. This spermatogonia initiates early meiosis, which results from premature induction of retinoic acid-response genes [19]. Collectively, these data suggest a direct involvement of AGO4 in mammalian meiosis and successful spermatogenesis, with analogous processes having been described in the nematode *C. elegans* and the fungus *N. crassa* [253].

A testicular tumor, which is the most common malignancy in young men, derives from abnormalities in germ cells during fetal development. Testicular Germ-Cell Tumor (TGCT) risk and testicular abnormalities in both parents-of-origin were found to be modulated by AGO2. AGO2, together with the RNA-binding protein A1CF, participate in the epigenetic control of germ-cell fate, urogenital development, and gamete function [60]. AGO2 is presented to be associated with Huntingtin (HTT), which is expressed, besides the human brain, in the testis. Both proteins are located inside P-granules, which are the centers of mRNA storage and degradation. Therefore, this provides an additional role to the well-known function(s) of the *HTT* gene [254,255].

The catalytic activity of AGO2 is also essential for successful oocyte maturation. Deletion of *AGO2* or *Dicer*, from the female germ-line, causes infertility due to defects during meiosis I [256]. Mice that express a catalytically inactive, knock-in allele of *AGO2* (*AGO2^ADH^*), exclusively in oocytes, present normal oogenesis and hormonal response, but impaired meiotic maturation with severe defects in spindle formation and chromosome alignment that lead to meiotic catastrophe [257]. AGOs may likely ensure proper centromere function and correct chromosome segregation, as chromosomal deficiencies are coupled with the AGO presence in the nucleus. This belief is reinforced by the ability of CSR-1, the *C. elegans* AGO-family homologous member, to associate with chromosomes [258]. In its absence, animals become infertile and demonstrate a multitude of meiotic and mitotic defects [259,260].

The implication of AGOs is not limited in spermatogenesis and oogenesis, but is expanded in the early stages of embryogenesis, as demonstrated in mouse models. Translational degradation of a proportion of maternal transcript messages engages the RNAi-machinery. *AGO2* and *AGO3* transcripts are both expressed in oocytes, while depletion of *AGO2* mRNA causes developmental arrest at the two-cell stage during the maternal-to-zygotic transition phase [259].

Thoroughly examining the AGO-expression profiles in gametes (sperms and oocytes) and abortion products may prove extremely important and very beneficial for infertile individuals, especially when no other clinical explanation is available.

## 3. Conclusions

AGO proteins are fundamental players in the post-transcriptional gene silencing process. Despite the large number of pioneering studies regarding their identified functions, there is still a gap to be filled in their importance to human pathologies. Emerging evidence for gene expression regulation and cell-fate determination via AGO(s) mediation has opened new windows for the mechanistic association of their functional-deregulation, gene-polymorphism, and protein-modification profiles with clinicο-pathological features leading to serious diseases such as viral infections, auto-immunities, cancers, metabolic dysfunctions, neuronal disorders, and infertilities. AGOs drive the mRNA-degradation machinery during viral infections by loading viRNA species. Anti-Su autoantibodies react to AGO2, which serves as their main target antigen during AIDs. AGOs are critically engaged in tumorigenesis-related pathways while they seem to regulate energy expenditure during the course of metabolism by loading specific sets of miRNAs. They act as key mitochondrial, translation initiation factors to facilitate ribosome-mRNA interactions. AGOs are responsible for the intact behavioral responses to chronic stress stimuli while critical mutations may cause autistic features and intellectual disabilities. Remarkably, AGO accumulation has been related to neuronal pathologies. The cleavage activity of AGO2 is essential for successful oocyte maturation. AGO4 ensures the apoptosis execution program during spermatogenesis. All the above clearly unveil the versatile and pivotal role of AGO in both physiological and pathological cellular settings. The reasonably raised question is if AGO abnormalities are the molecular causes of the previously mentioned pathologies and diseases, and, thereby, if they represent novel and powerful druggable therapeutic targets, or consequences of a global cellular deregulation and destabilization. Advanced investigation needs to be performed to fully clarify the implicated molecular mechanisms and to hopefully provide AGO-dependent therapeutic regimens for the successful management of human pathologies in the clinic.

## Figures and Tables

**Figure 1 ijms-21-04007-f001:**
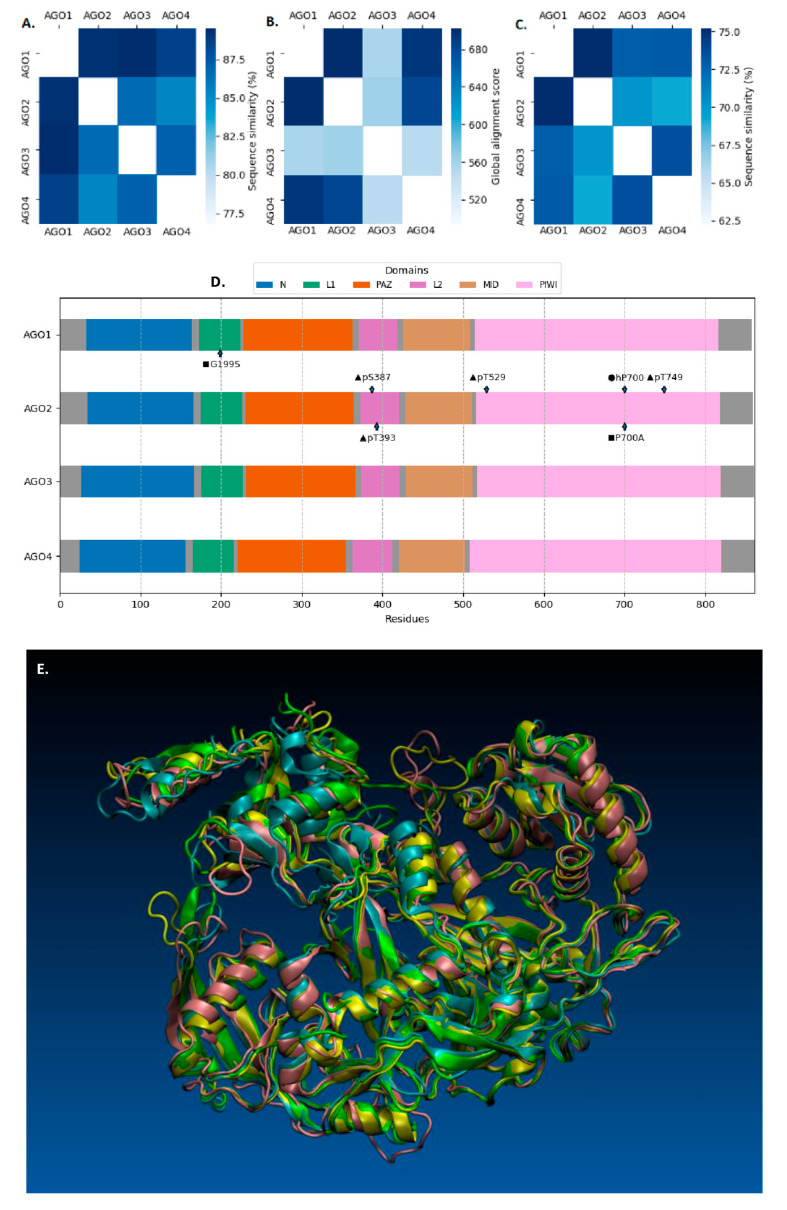
Comparative analysis of the human Argonaute (AGO) (AGO1–AGO4) protein-subfamily members. (**A**) Protein sequence similarity via global alignment. (**B**) Protein secondary structure via global alignment. (**C**) Coding-Sequence (CDS) similarity via global alignment. The alignments in (**A**) and (**C**) were yielded by the EMBL EMBOSS Needle with default settings [26]. The alignments in (**B**) were calculated using BioPython package with −2 as open gap penalty score, −1 as a gap extension penalty score, −1 as a mismatch score, and 1 as a matching score. Protein sequences were retrieved from ENSEMBL [27]. Protein-secondary structures (DSSP entries) were retrieved from RCSB PDB [28]. Plots were generated with Matplotlib and Seaborn Python packages. (**D**) Domains of AGO1–4 with annotations for mutations [■ = mutation]: G199S, P700A, and post-transcriptional modifications [● = hydroxylation (h), ▶ = phosphorylation (p)]: hP700, pS387, pT529, pT749, and pT393, respectively. Domain information was retrieved from Pfam database [29]. Plots were generated with Matplotlib and Seaborn Python packages. (**E**) A superposition of the AGO 1–4 tertiary structures in New Cartoon representation, as displayed by VMD software [30]. These structural models are refined PDB data from the PDB-REDO databank [31], 4KRE [AGO1, yellow colored] [20], 4Z4D [AGO2, cyan colored] [32], 5VM9 [AGO3, pink colored] [16], and 6OON [AGO4, green colored] [33]. The structural data were further processed using Schrödinger Maestro Suite (Schrödinger Release 2020-2, [34]) by removing existing water molecules and ligands and filling missing loops and side chains [Prime] [35], calculating the protonation states on physiological pH 7.4 [PROPKA] [36,37], and minimizing the free energy of the resulting structures with the OPLS3 force field [38].

**Figure 2 ijms-21-04007-f002:**
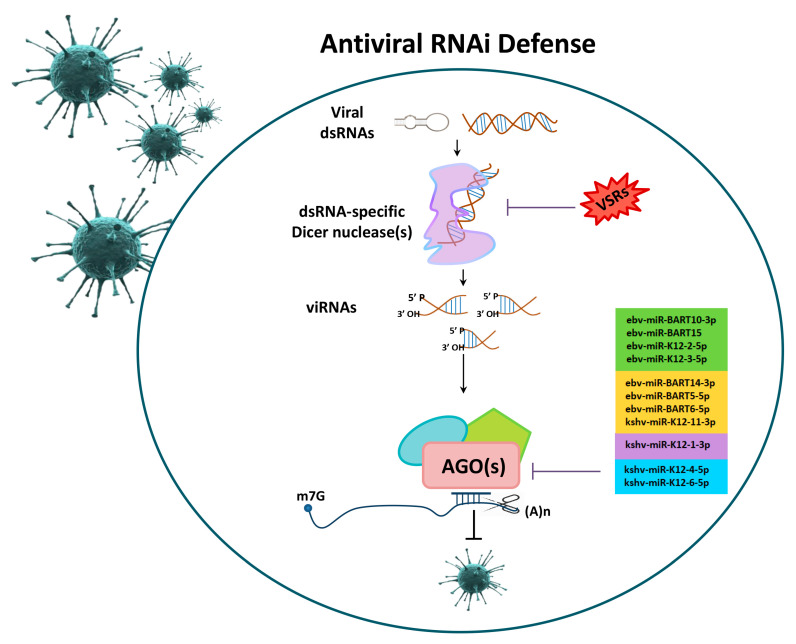
Post-transcriptional gene silencing during viral infections. Viruses that enter into eukaryotic hosts are able to produce their own miRNAs (viral miRNAs), which affects the immune system or aids their latency phase by reducing their expression. Viral double-stranded RNAs are cleaved by the dsRNA-specific Dicer nuclease, which is a discrete family of PRRs, to generate virus-derived small RNAs (viRNAs) and, hence, RNAi-silencing activation. To counterbalance viRNAs production, viruses express intrinsic Viral Suppressors of RNA silencing (VSRs) factors in order to protect themselves from Dicer-mediated cleavage and/or impede viRNAs loading onto AGO proteins. Based on the analysis of published datasets [82,83,84], human Ago-gene expression can be regulated by viRNAs derived from EBV and Kaposi’s Sarcoma-associated Herpes virus (KSHV).

**Figure 3 ijms-21-04007-f003:**
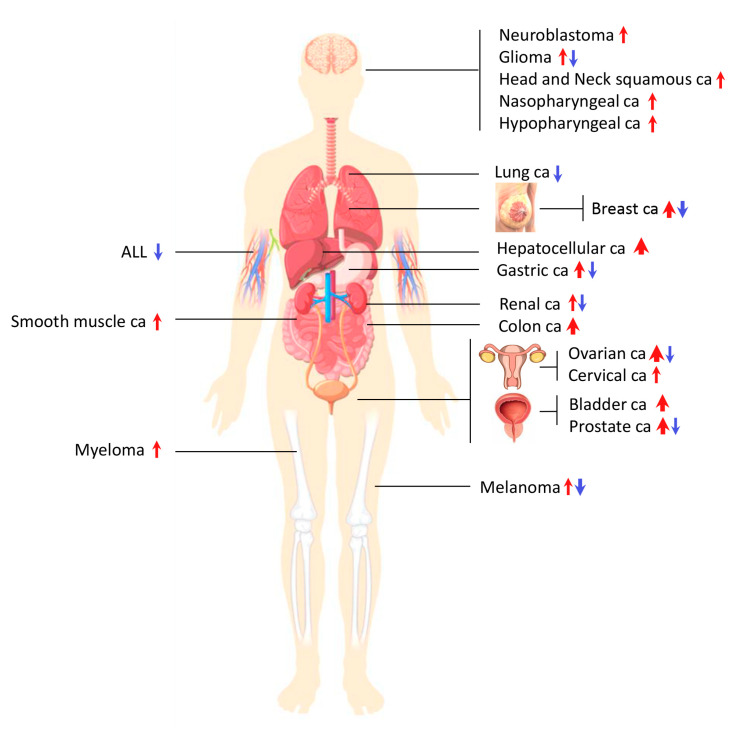
Deregulation of AGO-expression patterns in human cancer. The red and blue arrows indicate the increase and decrease of AGO-expression levels, respectively. The width of each arrow denotes the number of studies in which deregulated AGO contents were identified. Elevated AGO levels were reported in neuroblastoma, head and neck squamous, nasopharyngeal, hypopharyngeal, hepatocellular, colon, cervical, bladder, smooth muscle carcinomas, and myeloma. Reduced AGO levels were detected in lung adenocarcinoma and in acute lymphoblastic leukemia. Conflicting results, demonstrating both upregulation and downregulation of AGO expression, were described in breast, prostate, ovarian, gastric, and renal carcinomas as well as in glioma and melanoma.

**Figure 4 ijms-21-04007-f004:**
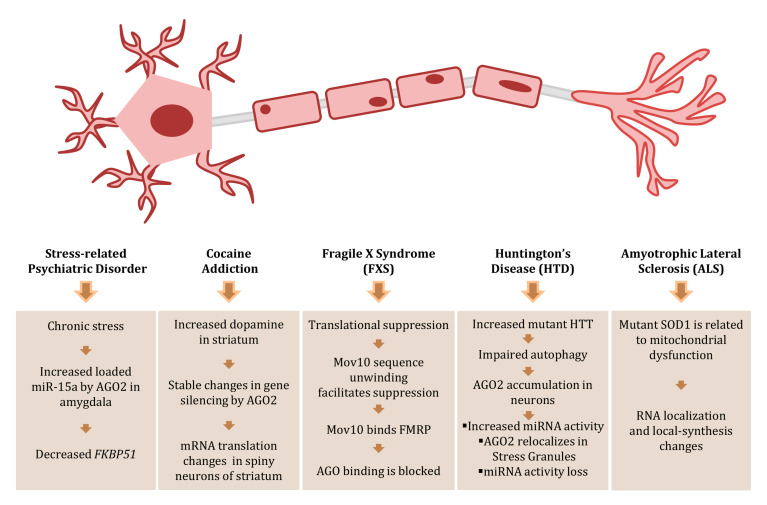
Argonaute (AGO) engagement in human neuropathology. Graphical overview of the AGO-associated pathologies in neurodegenerative diseases, such as ALS, HD, and in neuro-dependent diseases such as cocaine addiction and stress-related psychiatric disorders.

**Table 1 ijms-21-04007-t001:** AGO-deregulation profiling in human malignancies.

Cancer Type	AGO Type per Citation	Deregulation (Up or Down)	References
Gastric cancer	AGO2	upregulated	[141]
	AGO2	downregulated	[131]
	AGO2	upregulated	[142]
Prostate cancer	AGO2	upregulated	[141]
	AGO2	upregulated	[143]
	AGO2	downregulated	[130]
	AGO2	upregulated	[144]
Neuroblastoma	AGO2	upregulated	[141]
Bladder cancer	AGO2	upregulated	[129]
	AGO2	upregulated	[145]
	AGO1	upregulated	[146]
	AGO2	upregulated	
Myeloma	AGO2	upregulated	[147]
Breast cancer	AGO2	downregulated	[148]
	AGO2	upregulated	[126]
	AGO2	upregulated	[149]
	AGO2	downregulated	[148]
	AGO1 AGO2	upregulated	[150]
Ovarian cancer	AGO1 AGO2	upregulated	[151]
	AGO1 AGO2	upregulated	[152]
	AGO2	upregulated	[153]
	AGO2	downregulated	[154]
	AGO2	upregulated	[155]
Hepatocellular carcinoma	AGO2	upregulated	[156]
	AGO2	upregulated	[157]
	AGO2	upregulated	[158]
	AGO2	upregulated	[159]
Melanoma	AGO1	downregulated	[160]
	AGO2	downregulated	
	AGO3	downregulated	
	AGO4	downregulated	
	AGO2	downregulated	[127]
	AGO1 AGO2	upregulated	[161]
Cervical cancer	AGO2	upregulated	[162]
Renal carcinoma	AGO1 AGO2	upregulated	[163]
	AGO2	downregulated	[132]
Colon cancer	AGO2	upregulated	[142]
	AGO1	upregulated	[164]
	AGO2	upregulated	
	AGO3	upregulated	
	AGO4	upregulated	
	AGO2	upregulated	[165]
Nasopharyngeal carcinoma (NPC)	AGO2	upregulated (specific genetic variants)	[166]
Hypopharyngeal cancer	AGO2	upregulated	[167]
Lung cancer	AGO2	downregulated	[139]
Acute lymphoblastic leukemia (ALL)	AGO2	downregulated	[168]
Head and neck squamous cell carcinoma	AGO2	upregulated	[169]
Smooth muscle tumors	AGO2	upregulated	[170]
Glioma	AGO2	upregulated	[155]
	AGO2	downregulated	[154]

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
