# Peer review of "From the Argonauts Mythological Sailors to the Argonautes RNA-Silencing Navigators: Their Emerging Roles in Human-Cell Pathologies"

_ijms, 2020, doi:10.3390/ijms21114007_

Round 1
Reviewer 1 Report
The authors undertake the quite large task to review the role of AGO proteins not just in one particular but rather in all human diseases. They present a very thorough and balanced manuscript.
I only have the following comments/suggestions:
- Why was the particular order of subchapters chosen? The way it currently is presented, the significance of AGO contribution changes from each chapter to the next- I would maybe arrange it from weak to strong associations in disease pathologies.
- Along the same lines, I would suggest to add a table, in which all described diseases (or disease entities) are mentioned and ranked along the lines of “weak disease associations, strong disease associations, conflicting disease associations” (or similar).
- It is currently (in some parts of the manuscript) difficult to understand, whether the described disease associations are of direct AGO-associated nature or rather a consequence of global miRNA deregulation upon AGO level changes. Maybe another column could be added to the afore-mentioned table ranking the effects as direct vs indirect.
- Staying with the issue of direct vs indirect effects: the authors online mention one of the many CLIP (cross-linking and immunoprecipitation) papers (be it HITS-CLIP, PAR-CLIP etc.). Analyzing this papers in a bit more detail might prove interesting to at least elucidate the potential direct RNA-binding effects of AGO proteins.
- Would it be possible to present also 3D structures of the four AGO proteins (maybe in Figure) to underscore the similarity not only on amino acid level but also on a structural level between the four proteins?
- It would maybe also be nice to add tissue expression profiles of the four AGO proteins to visualize for the reader, in which tissues which AGO proteins are exclusively expressed and in which tissue their functions might overlap/complement one another.
- Line 52: there is a “not” missing in the sentence “… issues to be [not] easily resolved”.
- Line 102: it should be “existing” instead of “existed”.
- Line 107: I believe it should be “infertility”.
- Chapter 2.1: This quite long chapter presents in part quite complicated, partially opposing views, which relate only little to AGO itself- maybe it could be a bit shortened?
- Also, would it be possible to add a figure to the virus chapter to make it easier for the reader to grasp the information?
- Figure 3: It should only read “neuroblastoma” and “melanoma” (without the “ca”).
- Chapter 2.4: I would change the title to “AGOs in metabolic deficiencies” (or diseases).
Reviewer 2 Report
In this review, the authors highlight the role of Argonaute (AGOs) genes in humans, as well as highlighting the role of AGO genes both in in mammals and other organisms. In fact, AGO proteins are fundamental players in the post-transcriptional gene silencing process. Their role in human pathologies such as metabolic dysfunctions, viral infections, cancers, metabolic dysfunctions, infertilities and neuronal disorders, has yet to be clarified. But I think that this review is to be considered useful for this purpose.
Author Response
Dear reviewer,
Thank you very much for your support on our manuscript.